# Gestational Weight Gain as a Modifiable Risk Factor in Women with Extreme Pregestational BMI

**DOI:** 10.3390/nu17040736

**Published:** 2025-02-19

**Authors:** Chiara Lubrano, Federica Locati, Francesca Parisi, Gaia Maria Anelli, Manuela Wally Ossola, Irene Cetin

**Affiliations:** 1S.C. Ostetricia, Foundation IRCCS Ca’ Granda Ospedale Maggiore Policlinico, 20122 Milan, Italy; federica.locati@unimi.it (F.L.); francesca.parisi@unimi.it (F.P.); manuela.ossola@policlinico.mi.it (M.W.O.); irene.cetin@unimi.it (I.C.); 2Nutritional Sciences—Doctoral Programme (PhD), Department of Veterinary Medicine and Animal Sciences (DIVAS), University of Milan, 26900 Lodi, Italy; 3Department of Biomedical and Clinical Sciences, University of Milan, 20157 Milan, Italy; gaia.anelli@unimi.it; 4Department of Clinical and Community Sciences, University of Milan, 20122 Milan, Italy

**Keywords:** pregnant, obesity, inflammation, BMI, GWG, underweight women

## Abstract

The global rise in obesity presents serious concerns, particularly due to its association with pregnancy complications such as gestational diabetes, preeclampsia, cesarean delivery, and fetal macrosomia. Maternal obesity also contributes to intergenerational health risks, increasing the likelihood of long-term issues in offspring. Preconception counseling is an essential preventive measure to reduce complications; however, many women miss this opportunity due to unplanned pregnancies. This study explores the impact of pregestational body mass index (BMI) and gestational weight gain (GWG) on pregnancy outcomes, underscoring the importance of routine monitoring of these parameters. Existing studies identify both BMI and GWG as independent risk factors for adverse maternal and neonatal outcomes, with elevated BMI combined with excessive GWG posing an even greater risk. Specifically, a BMI > 30 kg/m^2^ doubles the risk of complications such as gestational diabetes, hypertension, and cesarean delivery. Additionally, a review of national and international guidelines highlights a lack of consensus on managing gestational weight gain in women with obesity, particularly regarding antepartum surveillance and timing of delivery. Similarly, no specific guidelines have been established for underweight pregnant women. Additionally, few studies have thoroughly assessed the maternal and fetal risks associated with underweight during pregnancy. Despite this, numerous studies have highlighted an increased risk of preterm birth (PTB) and small-for-gestational-age (SGA) infants. This narrative review emphasizes the need for further research to develop tailored guidelines for managing pregnant women based on pregestational BMI, ultimately improving maternal and neonatal health outcomes.

## 1. Introduction

Globally, the prevalence of obesity is escalating at an alarming rate, with significant short- and long-term implications for individual health, including an increased risk of cancer, diabetes, cardiovascular diseases, and chronic respiratory disorders [1], as well as for healthcare systems [2]. According to the World Health Organization (WHO), approximately 15% of women worldwide are classified as obese, with prevalence rates exceeding 30% in countries such as the United States and parts of Europe [1]. In Italy, around 9% of women are affected by obesity, marking a substantial increase over the past decade [3]. This trend is likely fueled by a complex interplay of factors, including increased availability of processed foods, sedentary lifestyles, and socioeconomic disparities, even within high-income countries.

As the prevalence of obesity continues to rise, its incidence among pregnant women has also increased [4], carrying significant implications for both maternal and neonatal health outcomes [5,6,7]. Maternal obesity is associated with an elevated risk of various pregnancy complications, including gestational diabetes, preeclampsia, venous thromboembolism, cesarean delivery, and fetal macrosomia [5,7]. Furthermore, maternal obesity has intergenerational effects [8,9], contributing substantially to the burden on public health systems; offspring of obese mothers are nearly five times more likely to develop obesity themselves [7], leading to well-documented adverse health outcomes.

For this reason, preconception counseling may be essential for implementing preventive strategies [10]. However, as most pregnancies are unplanned, a significant number of women miss the opportunity to benefit from preconception evaluations.

In this context, assessing GWG may serve as a simple, quick, and cost-effective measure, offering valuable indirect information about maternal energy intake and the nutritional status. GWG emerges as a potentially modifiable factor to reduce adverse pregnancy outcomes [11]. During pregnancy, increased energy intake is necessary to support the deposition of new maternal, placental, and fetal tissues, triglyceride storage in maternal fat tissue, and the heightened metabolic demands of both the mother and fetus. Therefore, achieving an appropriate energy balance—considering energy intake, expenditure, and storage in maternal and fetal tissues—is crucial.

The energy stored in fetal and placental tissues is relatively consistent among all women and has a minimal impact on overall energy requirements. However, energy needs vary significantly based on maternal fat mass, as fat serves as a high-density energy reserve. Women with a low BMI and limited fat stores require a higher energy intake, while those with a high BMI and greater fat stores may need little to no increase in energy intake, particularly during the first half of pregnancy, when maternal fat accumulation is most prominent [12].

Moreover, limited data are available on underweight women who become pregnant, and there is a lack of clinical guidelines for their management [13]. While numerous studies highlight maternal underweight as a risk factor for adverse pregnancy outcomes [14,15,16,17,18], the lower prevalence of this condition in the modern population has resulted in fewer studies and limited recommendations.

In conclusion, there is currently no consensus on the optimal management of obese or underweight women during pregnancy to reduce maternal and fetal complications. Key aspects lacking agreement include appropriate GWG, recommended evaluations during prenatal visits, and the timing of delivery. This results in a significant gap in clinical practice for healthcare providers and often leads to the application of different internal protocols across various birth centers. The objective of this narrative review is to examine the effect of pregestational Body Mass Index (BMI) and GWG on pregnancy outcomes and to explore their role in prenatal care decisions.

## 2. Materials and Methods

This review follows the PRISMA (Preferred Reporting Items for Systematic Reviews and Meta-Analyses) guidelines to ensure a comprehensive and structured synthesis of the available literature. The search was limited to studies published in English and within the last 10 years, except for studies of particular significance. The inclusion criteria for this review were studies that investigated outcomes in pregnant women based on pregestational BMI and GWG. Observational studies, reviews, meta-analyses, and randomized controlled trials were included. All identified studies were screened for year, citation, title, authors, and abstract.

Additionally, both international and national guidelines on the management of gestational weight gain in obese women during pregnancy were considered. The guidelines were identified using the search terms “Guidelines” AND “GWG” OR “obese women” OR “nutrition” OR “induction of labor”.

An extensive literature search was conducted using Pubmed, Google Scholar, and Scopus, employing the following search terms: “pregestational BMI” [All Fields] OR “gestational weight gain” [All Fields] OR “obese pregnant” [All Fields] OR “obese pregnant women” [All Fields] OR “underweight pregnant women” [All Fields] OR “underweight pregnant”, [All Fields] “DHA” OR “myo-inositol” OR “α-lipoic acid” AND obese pregnant” [All Fields], “DHA” OR “myo-inositol” OR “α-lipoic acid” AND “obesity” [All Fields], “physical activity” OR “diet” AND “obese pregnant” [All Fields], “inflammation” AND “obese” OR “obese pregnant”.

## 3. Results

### 3.1. Comparison of National and International Guidelines on GWG and Pregnancy Surveillance in Obese Pregnancies

After reviewing guidelines from major obstetrical societies, RCOG [19], FIGO [20], ACOG [21], SIGO [22], Polish Society of Gynecologists and Obstetricians [23], SOCG [24,25], RANZCOG [26] and NICE [27,28] provide recommendations for the management of obese women during pregnancy and delivery (Table 1). FIGO underlines that GWG is an independent risk factor for adverse health outcomes and that it has a cumulative effect when combined with pregestational BMI [20]. Consequently, FIGO recommends integrating GWG monitoring into routine antenatal care practices [20]. Lifestyle interventions that incorporate diet and physical activity during pregnancy have also demonstrated reductions in the risks of hypertension, cesarean delivery, and respiratory distress in neonates [20]. In contrast, RCOG highlights the lack of consensus on optimal GWG targets and suggests that, until more evidence becomes available, promoting a healthy diet may be more effective than setting specific weight gain targets [19]. NICE considers the evidence insufficient to support routine weight measurement throughout pregnancy, except when clinically necessary. While no optimal weight gain has been established, reference estimates of gestational weight gain (GWG) based on pre-pregnancy BMI can guide healthcare professionals [27]. However, NICE underlines the importance of BMI assessment at the first antenatal visit, as a higher BMI increases the risk of complications [27]. Likewise, excessive GWG is associated with a higher risk of gestational hypertension, gestational diabetes, and fetal macrosomia [27]. Furthermore, NICE highlights that dietary interventions in women with obesity may reduce the risk of gestational diabetes, though there is insufficient evidence to suggest a significant impact on cesarean birth, hypertensive disorders, or neonatal outcomes [27,28]. The same applies to physical activity alone or the combination of diet and exercise [27,28]. Although NICE states that intentional weight loss during pregnancy is not recommended due to potential risks to the baby, it provides no clear guidance on whether this advice should vary based on pre-pregnancy BMI [27]. Additionally, NICE finds no strong evidence that any specific balanced diet is more effective than others in managing weight gain during pregnancy [27,28].

SOCG and RANZCOG indicate assessing BMI before pregnancy or ideally within the first trimester and monitoring gestational weight gain throughout pregnancy [24,25,26]. Additionally, RANZCOG, in line with WHO, reports a lower BMI cut-off for Asian women and a recommended weight gain rate of 0.22 (0.17–0.27) kg/week during the second and third trimester [26].

ACOG emphasizes the importance of preventing excessive GWG but notes that some studies found no significant differences in preterm delivery, cesarean delivery, or macrosomia when comparing intervention (diet and lifestyle) and non-intervention groups [21]. Additionally, ACOG guidelines [29], as well as Italian recommendations [30], suggest no additional caloric intake during the first trimester. In the second trimester, they recommend an increase of 340–350 kcal/day, followed by approximately 452–460 kcal/day in the third trimester. However, neither set of recommendations consider variations in pregestational BMI.

In 2019, Most et al. proposed a new approach to caloric intake during pregnancy, emphasizing the varying energy requirements due to body fat serving as an energy reserve (Figure 1). Specifically, they highlighted that caloric intake during the first trimester should range between 50 and 150 kcal/day, with adjustments based on physical activity level. In contrast, for the second and third trimesters, caloric intake for obese women should be reduced by 165 kcal/day, while, for underweight women, it should increase to as much as 365 kcal/day [31].

The Polish Society of Gynecologists and Obstetricians recommend a reduction in pregestational BMI in women with obesity to significantly decrease the risk for maternal and fetal complications. Polish guidelines [23], SOGC [24,25], SIGO [22] and RANZCOG [26] all agree on recommending additional folic acid supplementation, though with varying dosages. Additionally, SOGC and RNZCOF further suggest aspirin prophylaxis [24,25,26]. Low GWG, a balanced diet, and moderate-intensity physical activity are recommended for obese pregnant women [22,23,24,25,26].

RANZCOG emphasizes that the timing and frequency of ultrasound should be determined on the overall clinical evaluation [26].

ACOG, RCOG, and FIGO underline the importance of increased fetal surveillance during the third trimester. ACOG provides specific timing recommendations based on pregestational BMI: for women with a BMI of 35.0 kg/m^2^–39.9 kg/m^2^, monitoring should begin weekly from 37 gestational weeks, while, for those with a BMI > 40 kg/m^2^, monitoring should start weekly from 34 weeks [21]. In cases of reduced fetal movements, FIGO recommends heightened fetal surveillance in the third trimester [20]. Additionally, Polish guidelines advise performing at least four ultrasound examinations in obese women [23].

According to FIGO, SOCG, RANZCOG, and RCOG, induction of labor should be considered at term in obese women [19,20,24,25,26]. Specifically, FIGO recommends considering induction at 41 + 0 weeks of gestation for women with a BMI ≥ 35 kg/m^2^ to reduce the risk of intrauterine death [20]. RCOG supports elective induction at 38, 39, or 40 weeks to reduce the odds of macrosomia and cesarean birth without increasing adverse outcomes [19]. Similarly, SOCG recommends induction at 39–40 gestational weeks for women with a BMI ≥ 40 kg/m^2^ [24,25]. RANZCOG suggests induction before their due date for women with BMI ≥ 50 kg/m^2^. Conversely, ACOG states that maternal obesity alone should not be an indication for induction [21]. Similarly, according to the SIGO, there is not sufficient evidence to recommend induction of labor at 39 weeks of gestation in cases of maternal obesity [32]. Finally, NICE suggests individual assessment for women with a BMI of 30–35 kg/m^2^ when planning the place of birth, while women with a BMI > 35 kg/m^2^ should be referred to an obstetric unit [33,34]. Additionally, ACOG recommends considering a longer first stage of labor before performing a cesarean delivery for labor arrest [21].

Given the increased risk of postpartum hemorrhage, RCOG, RANZCOG and FIGO, as well as Polish guidelines, recommend active management of the third stage of labor [19,20,23,26].

### 3.2. Body Mass Index and Weight Gain: A Dual Risk for Adverse Pregnancy Outcomes

Recent evidence indicates that both pregestational BMI and GWG are independent risk factors for maternal and neonatal complications [5,6,35,36]. Additionally, the combination of BMI > 30 kg/m^2^ and excessive GWG further increases the risk of pregnancy complications, with the risk stratified by obesity class [7,37]. However, assessing the impact of GWG and pregestational BMI on maternal and neonatal outcomes as distinct risk factors remains complex.

The LifeCycle Project, a meta-analysis that included 196,670 participants from 25 cohort studies across Europe and North America, demonstrated that the absolute risk of any adverse outcome increases from 31.7% to 54.4% in women with obesity compared to those with appropriate pregestational BMI [38]. Specifically, when comparing women with normal pregestational BMI and excessive GWG to those with obesity and appropriate GWG, the latter group exhibits double risk for severe complications: preeclampsia (6.9% vs. 2.3%), hypertensive disorders (8.0% vs. 3.3%), gestational diabetes (3.9% vs. 0.8%), preterm delivery (6.2% vs. 3.0%), and cesarean section (22.4% vs. 13.9%) [38]. However, the risk for small for gestational age (SGA) infants is comparable between the two groups, while the risk for large for gestational age (LGA) fetuses increases by 66% in the obese group [38]. Additionally, the risks of SGA and LGA significantly increase in women with class III obesity (BMI ≥ 40 kg/m^2^), along with the risk of diabetes, hypertension, preterm delivery, and cesarean section [7].

Therefore, the concept that pregestational BMI plays a crucial role in pregnancy outcomes must be complemented by the knowledge that excessive gestational weight gain is also a risk factor for diabetes, hypertension, macrosomia, cesarean section, and childhood obesity, albeit to a lesser extent. [39].

Moreover, the recommendations from the Institute of Medicine (IOM) and WHO do not differentiate GWG among obesity classes. In contrast, several recent studies suggest that a more accurate approach to GWG during pregnancy may be necessary, particularly for class 3 obesity, where lower weight gain or even weight loss may be safe and improve maternal and fetal outcomes without increasing the risk of SGA [7,37,38]. In 2020 Devlieger et al. recommended that patients with Class 1 obesity (BMI from 30 to <35 kg/m^2^) during pregnancy should gain no weight, while those with Class 2 (BMI from 35 to <40 kg/m^2^) and Class 3 (BMI ≥ 40 kg/m^2^) obesity should aim to lose 4 kg and 5 kg, respectively [40].

Additionally, Chiossi et al. observed that maternal pre-pregnancy BMI had a more significant impact on hypoxic morbidity than GWG. Specifically, the risk of developing hypoxic composite neonatal morbidity—defined as the occurrence of stillbirth, neonatal death, resuscitation at birth, NICU admission, intracranial hemorrhage, periventricular hemorrhage grade III and IV, neonatal seizures, necrotizing enterocolitis, meconium aspiration, non-invasive or invasive respiratory support, respiratory distress syndrome, or neonatal sepsis—increased with maternal pre-pregnancy BMI (obesity class 1: OR 1.34 [95% CI 1.24–1.45]; obesity class 3: OR 1.79 [95% CI 1.58–2.02]); excessive GWG: OR 1.08 [95% CI 1.02–1.14]) [35]. These findings are further confirmed by Grandfils et al., who noted a 22% increase in NICU admissions for neonates born by class III obese mothers with a significant reduction in Apgar scores at birth [7]. Considering the long-term consequences on offspring, pre-gestational obesity has a more significant effect than excessive GWG on the development of childhood obesity, cardiovascular disease, and neurodevelopment issues [5,41,42].

### 3.3. Evaluable Strategies for Interventions in Obesity and Excessive GWG

Obesity is characterized by chronic, low-level immune activation due to increased adipose tissue [43]. Fat tissue in obesity is deficient in adiponectin, an immune-modulating hormone, and enriched in leptin, which contributes to immune dysregulation [44]. This creates a pro-inflammatory environment with an accumulation of pro-inflammatory immune cells, while anti-inflammatory elements, such as T-helper cells and regulatory T cells, are notably reduced, further amplifying inflammation. This immune imbalance promotes the secretion of pro-inflammatory cytokines, including IL-6, TNF-α, and IFN-γ exacerbating systemic inflammation and contributing to oxidative stress [43].

These inflammatory processes lead to epigenetic modifications and alterations in gene expression, resulting in placental dysfunction [45], pregnancy complications [43], and long-term disorders in offspring [42,46]. Although interventions during pregnancy may have a limited impact on the maternal obesogenic environment, which develops over an extended period, recent metabolomic studies show an impact of gestational weight gain on maternal metabolism and fetal–neonatal growth [47]. Excessive GWG, recognized as an additional risk factor by most scientific societies, may serve as a key indirect tool for monitoring adherence to a healthy diet. Diet, physical activity, and lifestyle interventions are universally recommended throughout pregnancy [22,24,26,48,49,50], with physical activity proving most effective when combined with a balanced diet [51].

Given the pro-inflammatory state in obese women, some researchers have proposed using anti-inflammatory agents during pregnancy. Docosahexaenoic acid (DHA), an omega-3 fatty acid, reduces inflammation by inhibiting the NF-kB pathway and modulating MAPK signaling, which helps shift immune cells like macrophages towards an anti-inflammatory phenotype. Supplementation with 800 mg of DHA daily in the second half of pregnancy has shown potential in improving placental function, lowering inflammatory markers, and reducing adiposity in offspring [52]. Myo-inositol is a sugar alcohol that enhances insulin sensitivity and reduces oxidative stress, activating antioxidant pathways and neutralizing free radicals through compounds like glutathione. Studies suggest that 2 g of myo-inositol daily may lower the incidence of gestational diabetes and hypertensive disorders; although, evidence on other pregnancy outcomes remains inconclusive [53,54].

α-Lipoic acid (LA), an organosulfur compound with potent antioxidant effects, also inhibits NF-kB and reduces inflammatory cytokine production [55]. Supplementation of LA (200–800 mg daily) has been linked to reductions in body weight, BMI, waist circumference, and systemic inflammation [56], as well as improved cervical shortening in pregnancy [55]. Studies in animal models indicate that LA may help limit excessive maternal weight gain and improve glycemic control, highlighting its potential for maternal–fetal health [56]. 

### 3.4. The Impact of Maternal Underweight on Pregnancy Outcomes

While obesity is widely recognized as a major global health concern due to its increasing prevalence worldwide, including in Italy [13,57], significantly fewer studies have examined the impact of maternal underweight on pregnancy outcomes. The WHO defines underweight as a BMI below 18.5 kg/m^2^. Currently, the prevalence of underweight women of childbearing age is estimated to be around 3–6% both in Italy and globally [57,58]. Large cohort studies have shown that underweight pregnant women tend to be significantly younger, more likely smoking, single, with high level of education, and unemployed [59]. Maternal underweight has been associated with adverse pregnancy outcomes; however, comprehensive guidelines for managing underweight pregnant women are lacking.

A low pregestational BMI may be associated with underlying organic conditions, such as gastrointestinal diseases or chronic inflammation, as well as psychiatric disorders, eating disorders, and socioeconomic or lifestyle factors [59]. Therefore, during the initial assessment, treatable causes of low BMI should be excluded before diagnosing idiopathic underweight women [59]. Women with a very low BMI may benefit from nutritional counseling. Although not explicitly recommended by current guidelines, additional fetal and maternal surveillance may be considered if the BMI is below 18.5 kg/m^2^, particularly in cases of poor gestational weight gain.

A recent retrospective study involving 14,624 women, including 203 classified as underweight, examined the association between pregestational BMI and maternal–fetal outcomes. The results showed that underweight women had a 2.52-fold increased risk of having an infant with congenital anomalies (95% CI 1.12–5.64, *p* = 0.025) and a 1.88-fold increased risk of delivering a low birth weight infant (95% CI 1.27–2.79, *p* = 0.002). Additionally, they had 2.09 times higher odds of preterm birth (PTB) (95% CI 1.37–3.20, *p* = 0.001) [18]. Similarly, Lynch et al., in a retrospective cohort study of 581 underweight women, observed a significantly higher risk of preterm birth (OR: 2.4, 95% CI 1.4–4.2, *p* = 0.003) [16]. Additionally, a large retrospective cohort study in California analyzing 950,356 deliveries from 2007 to 2010, including 72,686 (7.6%) underweight women, found a correlation between increasing severity of underweight status and a higher risk of PTB. Specifically, the risk of preterm birth increased with the severity of underweight status. Women with a BMI of 17–18.49 kg/m^2^ had a 17% higher risk (RR = 1.17, 95% CI 1.14–1.21), those with a BMI of 16.00–16.99 kg/m^2^ had a 36% higher risk (RR = 1.36, 95% CI 1.28–1.45), and the severely underweight group (BMI < 16 kg/m^2^) faced a 54% higher risk (RR = 1.54, 95% CI 1.40–1.68) [15]. Mezzasalma et al. reported a higher incidence of congenital anomalies, particularly affecting the nervous system, orofacial structures, and the urogenital tract, in fetuses of underweight pregnant women [17].

Furthermore, a metanalysis confirmed an increased risk of low birth weight in neonates born to underweight women, reporting an adjusted relative risk of 1.64 (95% CI 1.38–1.94) compared to those born to women with a normal BMI [14]. Lefizelier et al. further supported this finding, observed a significantly lower average birth weight of 3055 ± 580 g in underweight women compared to 3280 ± 511 g in women with normal BMI (*p* < 0.001) [13]. While maternal obesity remains a primary focus in maternal health research, the impact of maternal underweight on pregnancy outcomes should not be overlooked [13].

### 3.5. Impact of Gestational Weight Gain on Pregnancy Outcomes in Underweight Women

According to IOM guidelines, women with a pre-pregnancy BMI < 18.5 kg/m^2^ should gain between 12.5 and 18 kg during pregnancy. Several studies have shown that gaining less than 8.0 kg in underweight women increases the risk of adverse outcomes [6,38,60,61], including higher odds of SGA infant, fetal growth restriction (FGR), and PTB. Specifically, GWG below the IOM recommendations in women with BMI < 18.5 kg/m^2^ is associated with an increased risk of SGA (OR 1.89, 95% CI: 1.67–2.14) and PTB (OR 2.41, 95% CI: 1.01–5.73) [38]. Conversely, GWG exceeding 18 kg in underweight women increases the risk of LGA infant (OR 2.17, 95% CI: 1.81–2.60) and cesarean delivery (OR 2.31, 95% CI: 1.62–3.29) [38]. Additionally, a study by Montvignier Monnet et al. not only confirms the increased risk of SGA and FGR in underweight women with inadequate weight gain, but also found a higher incidence of premature rupture of membranes (42% vs. 19%, *p* = 0.008) and anemia (50% vs. 31%, *p* = 0.026) in this group [61]. Despite these findings, maternal pre-pregnancy BMI appears to have a stronger impact on maternal and infant health outcomes than gestational weight gain, emphasizing the importance of nutritional and weight management strategies even before conception [6,38].

## 4. Discussion

Currently, guidelines reveal substantial variability in the management of obese patients, reflecting the need for stronger evidence. As obesity rates continue to rise globally, developing clear, evidence-based criteria will be vital for optimizing both maternal and fetal outcomes. Body mass index (BMI) remains a primary factor in maternal risk stratification of adverse maternal and neonatal outcomes. Therefore, promoting effective prevention and treatment strategies is essential to mitigate potential risks. Pre-conceptional counseling must always be recommended to all women of childbearing age, and health education programs will be increasingly necessary in the future. However, gestational weight gain (GWG) should be routinely assessed during pregnancy, as it provides essential information on pregnancy progression, maternal diet, and potential obstetric risks in both obese and normal-weight women. In addition, it is an inexpensive, quick, and accessible tool. Tailored physical activity and personalized nutritional programs should be recommended for all obese or underweight pregnant women. A multidisciplinary approach, involving obstetricians, nutritionists, exercise physiologists, and psychologists, could improve maternal and fetal outcomes.

Furthermore, weight loss after delivery and puerperium may represent important measures of reducing risk of developing cardiovascular disease or diabetes, particularly in high-risk women. In line with current guidelines, increased monitoring at term pregnancy may be considered for obese women, as it is for other high-risk conditions. Routine visits that include fetal ultrasound, blood pressure measurements, blood glucose monitoring, vitamin supplementations, and evaluation of gestational weight gain can be useful in reducing maternal and fetal adverse outcomes. Furthermore, more attention should also be paid to women with excessive GWG. However, monitoring strategies should be tailored based on individual risk factors to optimize care.

Further, prospective studies are necessary to investigate the use of anti-inflammatory molecules in obese women during the preconception period and pregnancy. Targeting inflammatory pathways could offer a different approach to improving placental function and reducing pregnancy complications. However, a healthy diet and lifestyle are fundamental to reducing maternal and neonatal risks. Consequently, exploring and discussing dietary habits with pregnant women during visits can significantly enhance pregnancy outcomes [20,49].

Given the elevated risk associated with major surgery in obese women, particularly the risk of thromboembolism, vaginal delivery should be prioritized as the preferred option, with clear communication of its risks and benefits. Notably, ACOG highlights the importance of allowing for a potentially longer first stage of labor in obese women [21]. Additionally, it may be necessary to distinguish between appropriate and excessive GWG across the various obesity classes. While existing guidelines offer general recommendations for GWG in obese women, future studies should explore whether specific targets for each obesity class could better align weight gain recommendations with minimized adverse outcomes. As our understanding of obesity’s impact on pregnancy evolves, personalized approaches that incorporate both maternal BMI and GWG will be key to enhancing outcomes for both mother and child.

Although the prevalence of underweight pregnant women is significantly lower than overweight and obese women, numerous studies have highlighted the associated risks, particularly preterm birth and SGA infants. Despite these well-documented adverse outcomes, there remains a notable lack of specific guidelines for managing underweight pregnancies. Given this gap, a personalized and multidisciplinary approach should be considered to optimize maternal and fetal health. A comprehensive assessment of underlying causes—including nutritional deficiencies, chronic diseases, and eating disorders—should be prioritized. Early identification of modifiable risk factors could allow targeted interventions, including nutritional counseling, dietary supplementation, and increased maternal–fetal surveillance in high-risk cases. Additionally, achieving appropriate gestational weight gain plays a pivotal role in reducing adverse pregnancy outcomes. During pregnancy, assessing cervical length and fetal growth may reduce the risk of preterm birth and SGA infants.

Future research should focus on establishing evidence-based management strategies for underweight pregnant women, particularly in cases of severe underweight or inadequate gestational weight gain. Large-scale prospective studies are needed to define dietary interventions and monitoring protocols for this population.

## Figures and Tables

**Figure 1 nutrients-17-00736-f001:**
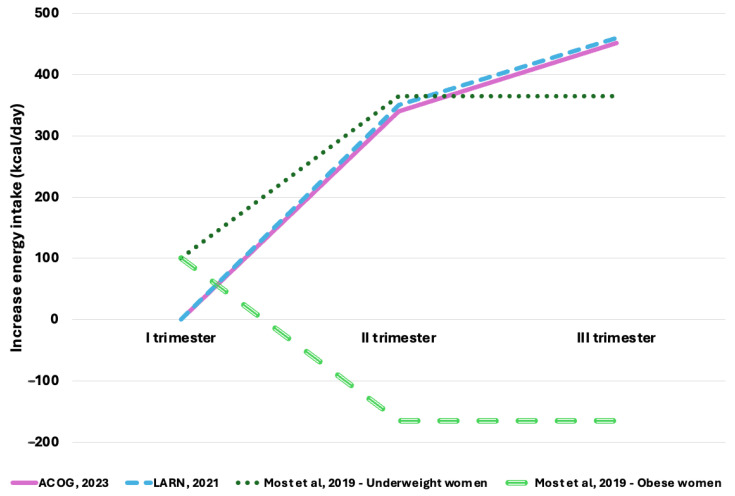
Comparison of the increase in energy intake recommendations during pregnancy according to ACOG (2023) [29], Italian guidelines (2021) [30], and Most et al. (2019) [31].

**Table 1 nutrients-17-00736-t001:** Comparison of guidelines from RCOG (2019) [19], FIGO (2020) [20], ACOG (2021) [21], NICE (2025) [27,28], SIGO (2018–2022) [22], RANZCOG (2022) [26] and the Polish Society of Gynecologists and Obstetricians (2023) [23].

	RCOG	FIGO	ACOG	NICE	SIGO	Polish Society	SOCG	RANZCOG
**GWG** **Monitoring**	No monitoring	Monitoring	Monitoring	No monitoring	Monitoring	Monitoring	Monitoring	Monitoring
**Diet to reduce maternal and fetal complications**	Effective	Effective	Not effective	Effective (only for gestational diabetes)	Effective	Effective	Effective	Effective
**Physical Activity to reduce maternal and fetal complications**	-	Effective	Not effective	Effective (only for gestational diabetes)	Effective	Effective	Effective	Effective
**Surveillance in Pregnancy**	Increased fetal surveillance during the third trimester	Increased fetal surveillance during the third trimester	**BMI > 35 kg/m^2^** weekly from 37 gestational weeks **BMI > 40 kg/m^2^** weekly from 34 weeks	-	**BMI > 30 kg/m^2^** Close maternal–fetal monitoring **BMI > 40 kg/m^2^** Maternal screening for cardiomyopathies	At least four ultrasounds.	-	Based on the clinical evaluation
**Induction for BMI**	Induction from 38 gestational weeks	**BMI ≥ 35 kg/m^2^** Induction at 41 gestational weeks	Not recommended	-	Not recommended	-	**BMI ≥ 40 kg/m^2^** Induction at 39–40 gestational weeks	**BMI ≥ 50 kg/m^2^** induction before their due date
**Third stage of Labore**	Active management	Active management	-	-	Active management	Active management	-	Active management

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
