# Peer review of "Gestational Weight Gain as a Modifiable Risk Factor in Women with Extreme Pregestational BMI"

_nutrients, 2025, doi:10.3390/nu17040736_

Round 1
Reviewer 1 Report
Comments and Suggestions for Authors
A very interesting idea for a review, the subject of gestational weight gain and recommendations for women with obesity, even though not a new one, is still underestimated.
However, I would suggest analyzing more recommendations considering obesity, pregnancy and weight gain, not only IFSO, ACOG and Italian. Excluding other national recommendations, e.g. British or Polish, narrows the perspective.
Author Response
Dear Reviewer,
Thank you for your positive comment.
We have incorporated additional national guidelines as you suggested.
Reviewer 2 Report
Comments and Suggestions for Authors
This is my review of your manuscript "Gestational weight gain as a modifiable risk factor in maternal obesity"
The manuscript is well-structured, providing a comprehensive review of gestational weight gain and maternal obesity.
The abstract is fine. You could strengthen it by adding specific findings like the proportion of adverse outcomes in obese women, etc.
In the introductions you should state the lack of scientific knowledge that you will address.
In the methodology you should add inclusion/exclusion criteria, publication timeframe, and databases used (other than PubMed and Scopus).
The results should include a table summarizing guidelines’ differences and provide with or without a structured table, quantitative comparisons between BMI classes.
Authors must provide recommendations on intervention strategies beyond general lifestyle modifications.
There are some minor grammatical and format errors that should be revised.
Author Response
Dear Reviewer,
Thank you for your positive comment.
-
The abstract is fine. You could strengthen it by adding specific findings, such as the proportion of adverse outcomes in obese women, etc.
We have attempted to improve the abstract by including additional data (lines 20–23).
- In the introduction, you should state the lack of scientific knowledge that you will address.
We have revised the introduction accordingly (lines 74–79).
-
In the methodology, you should add inclusion/exclusion criteria, the publication timeframe, and the databases used (in addition to PubMed and Scopus).
Thank you for the suggestion. We have added the inclusion and exclusion criteria, the databases used, and the relevant timeframe (lines 83–96).
- The results should include a table summarizing the diLerences among the guidelines and provide, with or without a structured table, quantitative comparisons between BMI classes.
Thank you for this suggestion. We believe it adds clarity to the section (Table 1).
-
Authors must provide recommendations on intervention strategies beyond general lifestyle modifications.
Thank you.
We have incorporated this aspect in lines 395–397.
-
There are some minor grammatical and formatting errors that should be revised.
We have carefully reviewed the manuscript to improve its clarity and correctness.
Reviewer 3 Report
Comments and Suggestions for Authors
This paper gives a narrative review of the risks to mother and child associated with maternal obesity, maternal low BMI and levels of gestational weight gain (GWG) in pregnant women.
This paper is well-written and requires minimal revision.
MINOR POINTS
Figure 1 could be improved by using different styles for the four lines so that the figure is still interpretable when reproduced in monochrome.
Line 235
“examinated” should read “examined”
Author Response
Dear Reviewer,
Thank you for your comment.
- Figure 1: We have modified the figure as recommended to ensure it remains interpretable in monochrome.
- Line 235: Thank you. We have corrected "examinated" to "examined."
Best regards